∂ | **Open Peer Review** | Mycology | Research Article

# Decoding the chemical language of *Suillus* fungi: genome mining and untargeted metabolomics uncover terpene chemical diversity

Sameer Mudbhari,[1,2] Lotus Lofgren,[3] Manasa R. Appidi,[1,2] Rytas Vilgalys,[3] Robert L. Hettich,[2] Paul E. Abraham[2]

**ABSTRACT** Ectomycorrhizal fungi establish mutually beneficial relationships with trees, trading nutrients for carbon. *Suillus* are ectomycorrhizal fungi that are critical to the health of boreal and temperate forest ecosystems. Comparative genomics has identified a high number of non-ribosomal peptide synthetase and terpene biosynthetic gene clusters (BGC) potentially involved in fungal competition and communication. However, the functionality of these BGCs is not known. This study employed co-culture techniques to activate BGC expression and then used metabolomics to investigate the diversity of metabolic products produced by three *Suillus* species (*Suillus hirtellus* EM16, *Suillus decipiens* EM49, and *Suillus cothurnatus* VC1858), core members of the pine microbiome. After 28 days of growth on solid media, liquid chromatography–tandem mass spectrometry identified a diverse range of extracellular metabolites (exometabolites) along the interaction zone between *Suillus* co-cultures. Prenol lipids were among the most abundant chemical classes. Out of the 62 unique terpene BGCs predicted by genome mining, 41 putative prenol lipids (includes 37 putative terpenes) were identified across the three *Suillus* species using metabolomics. Notably, some terpenes were significantly more abundant in co-culture conditions. For example, we identified a metabolite matching to isomers isopimaric acid, sandaracopimaric acid, and abietic acid, which can be found in pine resin and play important roles in host defense mechanisms and *Suillus* spore germination. This research highlights the importance of combining genomics and metabolomics to advance our understanding of the chemical diversity underpinning fungal signaling and communication.

**IMPORTANCE** Using a combination of genomics and metabolomics, this study's findings offer new insights into the chemical diversity of *Suillus* fungi, which serve a critical role in forest ecosystems.

**KEYWORDS** fungi, ectomycorrhizae, *Suillus*, genome mining, metabolomics, secondary metabolites, terpene

Ectomycorrhizal (ECM) fungi are important community members in temperate and boreal forest ecosystems, where they form an obligate symbiosis with woody plant species. Ectomycorrhizal fungi trade fungal-scavenged nutrients, such as nitrogen and phosphorus, for host-derived photosynthetically fixed carbon and play critical roles in biogeochemical cycling (1). Fungi in the genus *Suillus* are important ECM symbionts that associate almost exclusively with host trees in the family Pinaceae (2), where they facilitate improved seedling establishment, drought resistance via improved water conductance, and ecological remediation of heavy metal-contaminated sites (3–5). In their role as essential root symbionts, ECM fungi interact with complex consortia of other organisms including plants, bacteria, viruses, invertebrates, and other fungi. Interactions

Address correspondence to Paul E. Abraham, abrahampe@ornl.gov.

The authors declare no conflict of interest.

See the funding table on p. 12.

between ECM fungi and these co-occurring community members are mediated by complex chemical signals including small molecules, proteins, and secondary metabolites (6). Despite the importance of these interactions, our understanding of the chemical diversity facilitating interactions between ECM fungi, their hosts, and the rhizosphere community is limited. Characterizing the identity of the metabolites involved in these interactions and the conditions under which they are expressed are the first steps in elucidating the diverse ecological functions of secondary compounds in environmental sensing, competition, and communication.

Unlike primary metabolites, secondary metabolites are not essential to cellular functions but often play critical roles in signaling, communication, and regulating inter- and intraspecies interactions between fungi and their surrounding communities (7). To date, most studies of fungal secondary metabolic diversity have focused on pathogens, saprophytes, and endophytes, particularly those in the phylum Ascomycota (8). This hinders our ability to appreciate the vast repertoire of secondary metabolites produced by fungi occupying diverse lifestyles. Previous studies of *Suillus* metabolomes have mostly focused on fruit bodies (mushrooms) (9–11) and were able to identify metabolites such as prenylated phenols and boviquinones (10). The exometabolomes of *Suillus*, on the other hand, have not been well characterized. Previous work using comparative genomics and genome mining-based predictions of Biosynthetic Gene Clusters (BGCs) in *Suillus* indicated that the genus may have a significantly higher capacity to produce terpenes and non-ribosomal peptides than other ECM genera (12). However, genome-mining-based approaches are unable to characterize most of these compounds past these broad metabolite classes, and the ecological roles and conditions necessary for their expression are unknown. Further, BGCs are often not expressed under standard laboratory conditions and require induction by altering environmental conditions (13). Techniques, such as OSMAC (one strain many compounds), have been successfully used to activate silent BGCs by systematically altering biotic and abiotic environmental variables (14). Today, it is widely recognized that one of the most efficient and effective forms of metabolite induction is coculturing fungi with other organisms (15).

Unlike many OSMAC strategies, coculture has the added benefit of being able to address ecologically relevant scenarios, including interactions between naturally co-occurring or co-evolving community members. As a first step toward characterizing the chemical diversity of *Suillus*, we chose three genome-sequenced species of *Suillus* known to co-occur and associate with the same species of host tree (*Pinus taeda*—loblolly pine). Genome mining was used to predict and study the similarity between secondary metabolite-producing BGCs across the three species. The three species were grown in monoculture as well as coculture for all pairwise combinations. Untargeted metabolomics was then used to characterize the exometabolites produced at the growth interface between two fungal cultures.

## MATERIALS AND METHODS

### Genome mining

Genomes for the three species of *Suillus* used in this study, *S. decipiens* EM49, *S. cothurnatus* VC1858, and *S. hirtellus* EM16, were first published and characterized in (12) and are publicly available from the JGI MycoCosm database (16). Biosynthetic gene clusters were predicted using antiSMASH v.6.0.1 (17), with the parameters (--taxon fungi –cb-general –cb-subclusters –cb-knownclusters –p fam2go). Orthology and conservation predictions between BGC were carried out via BiG-SCAPE with default parameters (18). The inputs for BiG-SCAPE were the GenBank files obtained from antiSMASH. The antiSMASH and BiG-SCAPE result files are provided in MassIVE (see Data Availability).

### Co-culture and growth assay

The three species of *Suillus* used in this study originally came from fruitbodies growing under *Pinus* species. The three species of *Suillus* were inoculated onto 100-mm

Petri dishes containing solid high carbon Pachlewski's media (19). Each Petri dish was inoculated with $n = 2$ 4-mm plugs placed exactly 2 cm from one another and equidistant to a diameter line intersecting the plate. Treatments included all pairwise combinations of the three *Suillus* species (at $n = 5$ biological replicates). Single-species controls were inoculated with two plugs as above ($n = 5$ biological replicates). Therefore, altogether five biological replicates were used for each sample group. The media-only samples were used as a negative control ($n = 3$). Cultures were grown for 28 days, in the dark, at room temperature. Starting at 7 days post inoculation (dpi), the colony area was measured twice per week using background illumination and outlining the colony margins on the bottom side of each Petri dish using a fine-tip marker. At the end of the 28-day growth period, we captured images of the bottom of each Petri dish using a desktop scanner and calculated the colony area at each time point using the program imageJ (20). After 28 days, three agar plugs were collected, pooled together, and placed in 1.5-mL cryotubes using a sterile brass core borer, taking plugs from along the diameter line of the plate and capturing the interaction zone between the two cultures. After collection, samples were immediately frozen in liquid nitrogen and stored at −80°C until sample processing.

## Metabolomics sample preparation

The frozen agar plugs containing mycelia were lyophilized using a Labconco Freezone freeze dryer (Labconco Equipment Co., KS, USA) until completely dry. The freeze-dried agar plugs were then processed using a biphasic extraction method by mixing 0.5 mL of cold liquid chromatography–mass spectrometry (LC-MS)-grade water with 0.5 mL of cold hydrated ethyl acetate, vortexed for 1 min and then kept at 4°C overnight for extraction. The ethyl acetate and water fractions were then separated by aspiration. For the aqueous fraction, samples were filtered using a 10-kDa filter (Sartorius Vivaspin 2 Centrifugal Concentrator Polyethersulfone) by centrifugation at $4,500 \times g$ to remove the remaining agar particulates. After filtering, the aqueous extract was freeze dried and resuspended in an aqueous solvent (5% acetonitrile, 0.1% formic acid), while the ethyl acetate extract was air dried in a chemical fume hood until dry and then resuspended in an organic solvent (70% acetonitrile, 0.1% formic acid). All samples were stored short term at 4°C until liquid chromatography-electrospray ionization tandem mass spectrometry (LC-ESI-MS/MS) measurements.

## Liquid chromatography-electrospray ionization tandem mass spectrometry (LC-ESI-MS/MS)

All samples were analyzed using ultra-high-pressure liquid chromatography coupled with a ThermoFisher Q-Exactive Plus mass spectrometer. For each sample, 10 μL was injected and allowed to flow across an in-house-constructed nanospray analytical column (75 μm × 150 mm) packed with a 1.7-μm C18 Kinetex RP C18 resin (Phenomenex). The mobile phase included solvent A (95% water, 5% acetonitrile, 0.1% formic acid) and solvent B (70% acetonitrile, 30% water, 0.1% formic acid). The metabolites were separated across a 30-min linear organic gradient (250 nL/min flow rate) from 5% aqueous solvent (5% acetonitrile, 0.1% formic acid) to 100% organic solvent (70% acetonitrile, 0.1% formic acid). All MS data were acquired by Xcalibur software version 4.3 using the top N method where N could be up to 5. Target values for the full-scan MS spectra were $3 \times 10^6$ charges in the 135–2,000 *m/z* range with a maximum injection time of 100 ms. Transient times corresponding to a resolution of 70,000 at *m/z* 200 were chosen, and a 2.0 *m/z* isolation window and an isolation offset of 0.5 *m/z* were used. Fragmentation of precursor ions was performed by stepped higher-energy C-trap dissociation with normalized collision energies of 10, 20, and 40 eVs. MS/MS scans were performed at a resolution of 17,500 at *m/z* 200 with an ion target value of $1.6 \times 10^5$ and a 50-ms maximum injection time. Dynamic exclusion was set to 10 s to avoid oversampling of abundant metabolites. A more detailed listing of the parameters can be found in Supplemental File 1. The identification and quantification analyses for untargeted LC-MS/MS data were performed using Thermo Scientific Compound Discoverer (CD)

v3.3.1, MetaboAnalyst v6.0 (21, 22), GNPS (Global Natural Products Social Molecular Networking) (23, 24), and Skyline v22.2 (25, 26).

## Data processing for LC-ESI-MS/MS

The .raw files obtained from the measurements were analyzed using Compound Discoverer v3.3.1 using the untargeted metabolomics workflow. For this, the following workflow steps were used: first, input .raw files were processed by the "select spectra" node using the default settings (every spectrum from 0 to 30 min was used). The default option of a signal/noise (S/N) ratio of 1.5 was used for peak filtering. For the "detect compound" node, again, the default setting from the untargeted metabolomics workflow was used, including a mass tolerance of 5 ppm, a minimum peak intensity of 10,000, a minimum number of scans per peak equal to 5, and a most intense isotope set as "True". The compound detection was set to ions [M + 2H], [M + 3H], and [M + H] and base ions as [M + H]. Then, the "group compound" node was selected in the processing workflow where the settings included mass tolerance set to 5 ppm and retention tolerance in minutes of 0.02. The preferred ion was set at [M + H], and area integration was done for the most common ion. Similarly, the peak rating contributions were also included in the workflow. The peak rating filter was applied such that only peaks crossing the peak rating threshold of 4 in at least two files were retained. The "fill gaps" node from the processing node was used with the following settings: mass tolerance equal to 5 ppm, signal-to-noise ratio threshold of 1.5, and real-time peak detection and restrictive gap filling set to "True". The node "mzVault" was selected for spectral matching against the high-resolution NIST 2020 spectral library and the mzCloud spectral library. The nodes "predict composition" and "ChemSpider" were selected to further annotate compounds based on precursor mass. Then, the "mzLogic" node was selected with FT fragment mass tolerance set to 10 ppm, IT fragment mass tolerance set to 0.4 Da, maximum number of compounds set to 0, the maximum number of mzCloud similarity results to consider per compounds set to 10, and a match factor threshold set to 30. Next, the "assign compound annotations" node in the workflow was used to annotate in the following order: mzVault (NIST2020), mzCloud, predicted compositions, Chemspider, and Metabolika Search. The scoring rule included use of mzLogic set to "True", the use of spectral distance set to "True", The SFit threshold set to 20, and SFit Range set to 20. Finally, the differential analysis node was selected with the Log10 transformed value set to "True."

The resulting data matrix was processed further to obtain a non-redundant list of putative metabolite identifications. First, the compound identifications were filtered to only retain those that had a confident spectral match to either mzCloud or mzVault spectral libraries. Next, the data matrix was filtered to remove compound identifications that were indistinguishable by annotated molecular weight and retention time to obtain a non-redundant list of putative metabolites.

For data post-processing, the data matrix of the resulting non-redundant putative metabolite identifications and their associated quantitative values were analyzed by MetaboAnalyst v6.0. To identify metabolites with relative abundances greater than those observed in media controls, compound peak areas were Log10 transformed, and a Student's t-test was performed. The metabolites with Log2 fold change greater than 2 and an FDR-adjusted P-value less than 0.05 were selected.

For the GNPS-based molecular networking analysis, .raw files were first converted to .mzML format using the software MSConvert (27, 28). The .mzML files were transferred to the MassIVE server using the software WinSCP (https://winscp.net/). Then, using the GNPS software, the classical molecular networking workflow was used to process the provided .mzML files. For this workflow, the precursor ion mass tolerance was set at 0.01 Da, and fragment ion mass tolerance was set at 0.01 Da. The results were then subset to include only those metabolite feature nodes with a cosine similarity score of 0.7 or greater and with at least six matched fragment ions. The maximum size of each subnetwork was set to 100.

Skyline v22.2 was use for manual curation of putative metabolite relative abundances in this study. That is, retention time boundaries per metabolite were manually refined across all the data files. Skyline results are provided in MassIVE (see Data Availability).

## RESULTS

### Fungal growth was marginally different between inter- and intraspecies pairings

Colony area increased with incubation time for all the fungal cocultures starting from 7 days post inoculation to day 28 when samples were collected, indicating that all cultures were actively growing when agar plugs were collected for exometabolomic analysis. Generally, isolates grown in intraspecies pairings reached larger colony areas than isolates grown in interspecies (coculture) pairings, but none of these differences were statistically significant (ANOVA, $P > 0.05$) (Fig. S1).

### Genome mining and analysis of biosynthetic diversity predicts specific types of secondary metabolites

Using antiSMASH v6.0.1, we predicted the total number of BGCs for each of the genomes. *S. cothurnatus* VC1858 had the highest predicted number of putative BGCs (53), whereas *S. hirtellus* EM16 and *S. decipiens* EM49 had a lower number of BGCs (43 and 36, respectively) (Fig. 1A). In general, the total number of predicted BGCs was related to genome size for all three species. As found previously (12), the number of BGCs with genes for non-ribosomal peptide synthetase-like (NRPS-like) proteins and terpene synthases was much higher in all three genomes compared to that of BGCs representing other classes of backbone enzymes, such as polyketide synthases (PKSs) or hybrid BGCs with multiple types of backbones. In total, we found 38 terpene synthase BGCs in *S. cothurnatus* VC1858, 23 in *S. hirtellus* EM16, and 19 in *S. decipiens* EM49. Similarly, we identified 14 NRPS-like genes in *S. hirtellus* EM16, 12 in *S. cothurnatus* VC1858, and 14 in *S. decipiens* (Fig. 1B). This new comparative analysis of BGCs suggests that there is variation in secondary metabolite production across *Suillus* strains.

BiG-SCAPE was used to construct sequence similarity networks between BGCs to identify gene cluster families across the three *Suillus* genomes. This analysis revealed that the BGCs containing terpene synthases and NRPS-like genes are remarkably diverse (Fig. 1C and D), suggesting there is a vast chemical space to be discovered. Overall, 55 predicted BGCs with terpene synthases did not cluster into gene families based on sequence similarity, four were found to be orthologous across the three species of *Suillus*, two were orthologous between *S. hirtellus* EM16 and *S. cothurnatus*, and one was orthologous between *S. cothurnatus* VC1858 and *S. decipiens* EM49. Regarding BGCs containing NRPS-like genes, 23 did not cluster based on sequence similarity, four were orthologous between the three species, two were orthologous between *S. cothurnatus* VC1858 and *S. decipiens* EM49, and only one orthologous cluster was observed between *S. hirtellus* EM16 and *S. decipiens* EM49.

### Diverse classes of metabolites were identified from the exudates of *Suillus* depending on monoculture vs coculture conditions

The untargeted metabolomics data-processing workflow in Compound Discoverer 3.3.1 was used for spectral matching. Overall, this workflow predicted chemical formulas for 42,933 compound features that have a measured retention time and a relative abundance based on a calculated peak area (Table S1). Among these compound features, a metabolite annotation could be assigned to 19,400 features based on accurate precursor mass alone using the ChemSpider database or with further confidence by matching measured fragmentation spectra to spectral libraries in both the mzCloud database (29) and the high-resolution NIST 2020 database (30). Next, data were filtered to only consider compound features matching either the mzCloud and/or NIST 2020 databases, resulting in a list of 3,769 putative compound identifications (Table S2). Principal

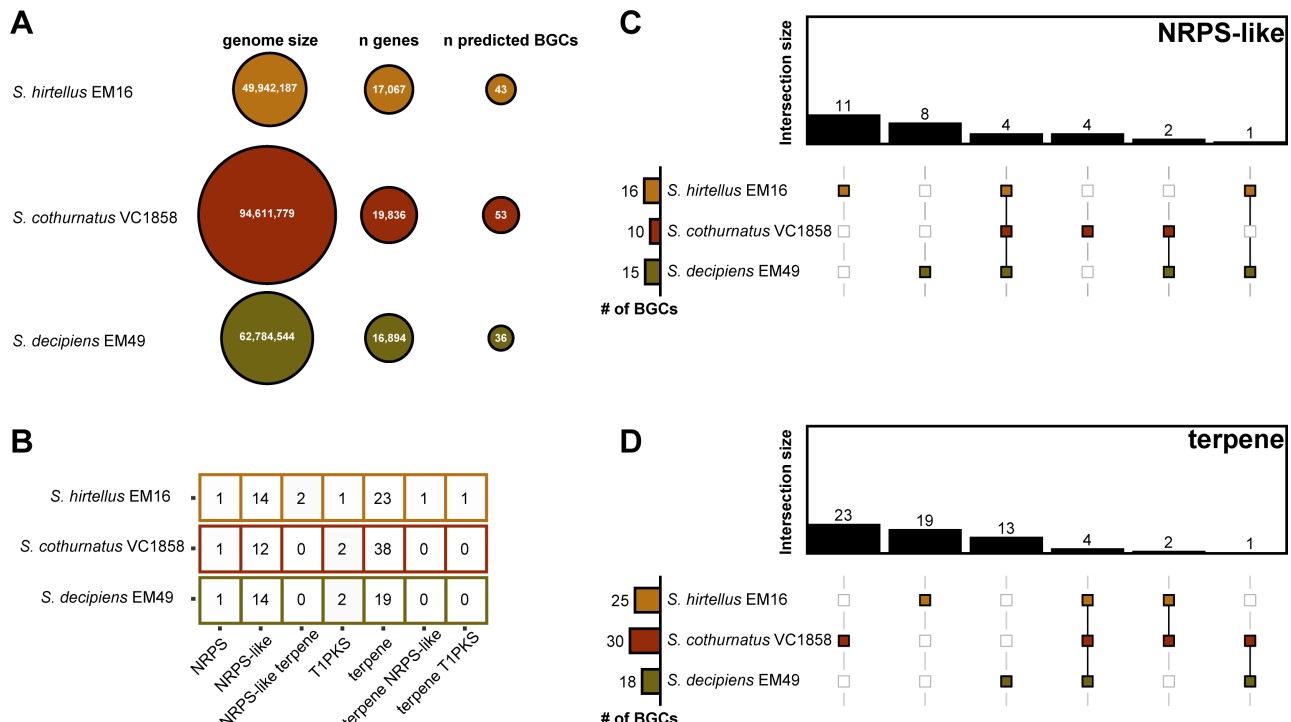

FIG 1 Genome mining revealed diversity of compounds encoded in three species of *Suillus*. (A) Genome size, number of genes, and predicted BGCs vary across *S. cothurnatus* VC 1858, *S. hirtellus* EM16, and *S. decipiens* EM49. (B) The tool antiSMASH v.6.0.1 detected and characterized the number of BGCs in the genome of *S. cothurnatus* VC1858, *S. hirtellus* EM16, and *S. decipiens* EM49. The tool BiG-SCAPE showed low grouping between architecture and sequence similarity for (C) NRPS-like and (D) terpene-classified BGCs.

component analysis was performed separately for the aqueous and organic fractions using MetaboAnalyst version 6.0 (21), with the resulting plots displaying discrete grouping between biological replicates and separation between coculture groups (Fig. 2A and B). Further manual data curation was performed to identify a non-redundant list of compound identifications that matched with mzCloud and/or the NIST 2020 high-resolution library resulting in a list of 770 putative metabolites. That is, the data were filtered to remove compound identifications that were indistinguishable by annotated molecular weight and retention time (i.e., isomeric) to obtain a non-redundant list of putative metabolites. The data set was then filtered to compound features with abundances significantly greater in sample groups when compared against the media blank control (adj. *P*-value < 0.05 and a Log2 fold change >2) resulting in a final list of 487 putative metabolites (Table S3). ClassyFire version 1.0 (31) was used for chemical taxonomy classification (Fig. 3), and there were 85 chemical classes observed for 467 out of 487 metabolites (Table S4). "Benzene and substituted derivatives" and "carboxylic acids and derivatives" were the most abundant chemical classes with 58 metabolites. This was followed by prenol lipids with 33 metabolites. Importantly, prenol lipids encompassed 8 different prenol lipid subclasses among which terpene lactones (10), diterpenoids (8), and sesquiterpenoids (5) were the top three most abundant subclasses.

To help define the conditions under which a particular metabolite or class of compounds are induced, we assessed the degree of overlap in putative identifications observed in the intra- and interspecies cocultures. An UpSet plot was created separately for compound identifications detected in an aqueous or organic fraction sample (Fig. 2C and D). In general, both UpSet plots showed that a relatively higher number of putative metabolites were observed in coculture conditions when compared to monoculture conditions. In the aqueous fraction, among the interspecies pairings, the highest number of unique features were observed among the S. *hirtellus* EM16–S. *cothurnatus* VC1858 (7).

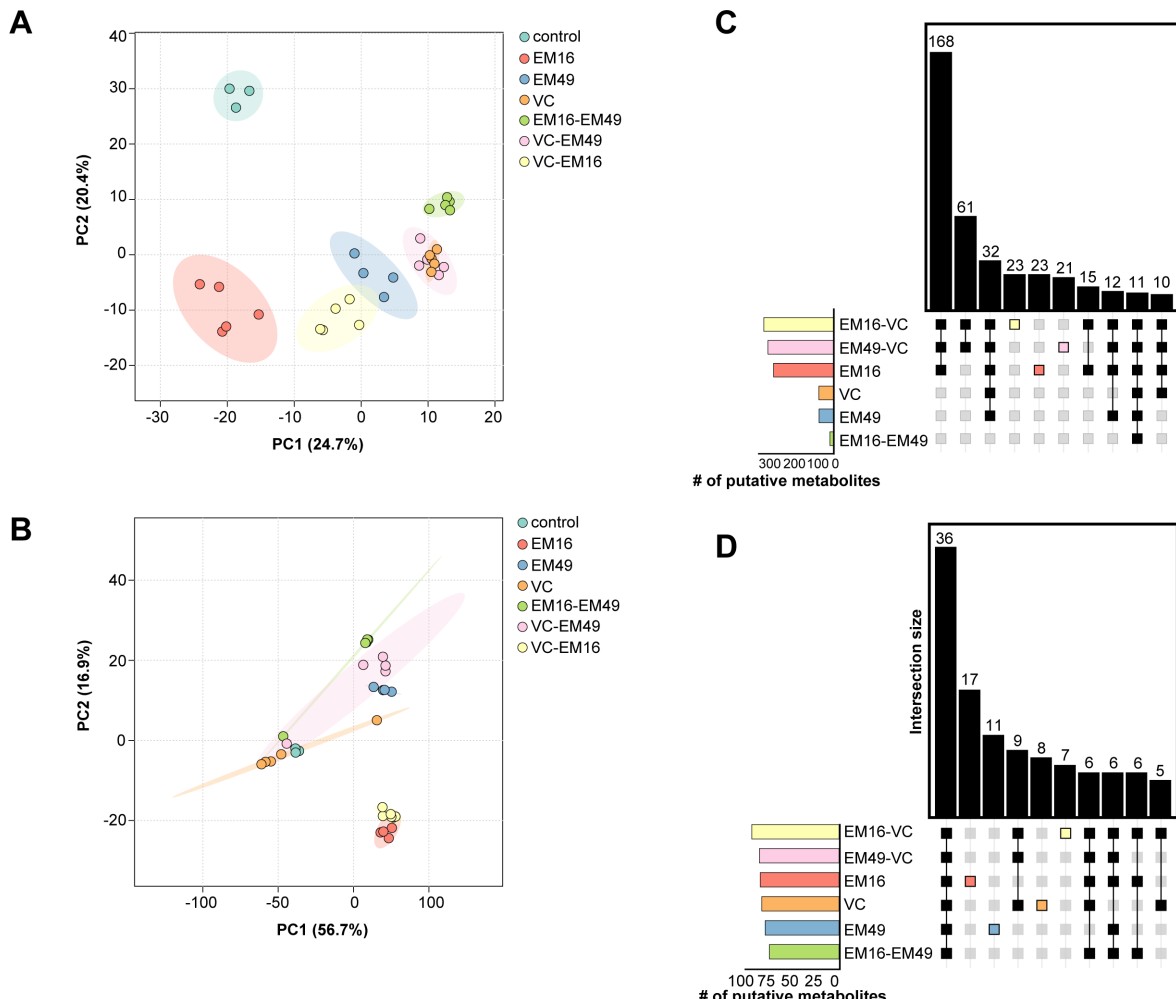

**FIG 2** Principal component analysis (PCA) plot and UpSet plot showed metabolite variation among different culture conditions. PCA for metabolites observed in (A) ethylacetate fraction and (B) aqueous fraction showed clear separation among culture conditions. The replicates are closer together, while culture groups are well separated along the PCA space. UpSet plot showing the number of putative metabolites that are present in either multiple-culture condition or present in only a specific-culture group for (C) ethylacetate and (D) aqueous fractions. Only a subset of the UpSet plot is shown to show the number of metabolites shared among different sample groups.

Similarly, for the organic fraction, the *S. hirtellus* EM16–*S. cothurnatus* VC1858 interspecies pairing produced the highest number of unique metabolites (23).

Next, we were interested in determining which coculture conditions produced the greatest diversity of chemical classes. In total, 59 putatively identified metabolites were present only in coculture conditions. ClassyFire-based classification of these metabolites resulted in 25 different classes, and we assessed the overlap of these classes across each sample. Across both fractions, the greatest representation of chemical diversity was observed for the S. *hirtellus* EM16–*S. cothurnatus* VC1858 interspecies pairing (Fig. S2 and S3). In general, the most represented chemical classes were "benzene and substituted derivatives" (11), "carboxylic acids and derivatives" (7), and "prenol lipids" (4).

## The exometabolome of *Suillus* was found to be rich in terpenes, which aligns well with genome mining-based prediction of terpene-coding potential

A relatively large number of putative terpenes were identified using untargeted metabolomics based on MS/MS similarity to reference spectra present in the high-resolution NIST 2020 and mzCloud libraries. Overall, 41 different terpenes had a high confidence match to a reference spectra, and analysis of taxonomic subclass showed eight different

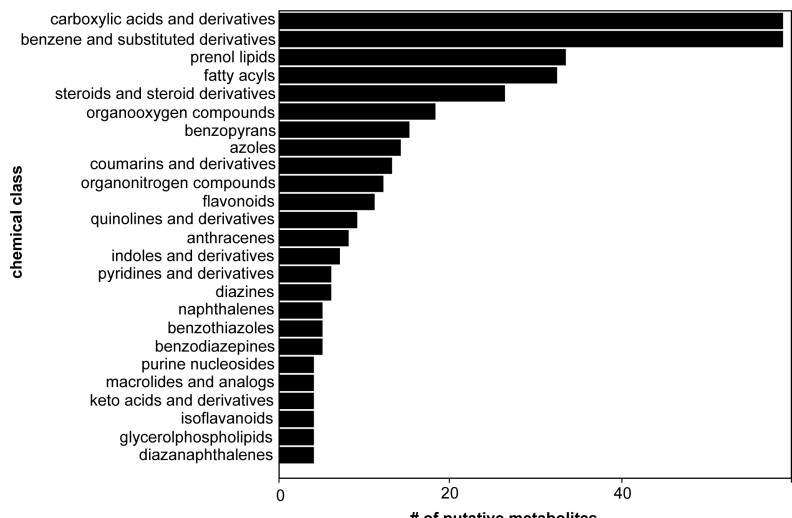

**FIG 3** Compound classification highlighted chemical diversity of the putatively identified metabolites. The tool ClassyFire classified the chemical taxonomy of compounds for the non-redundant list of 487 compounds identified using Compound Discoverer workflow. This analysis showed carboxylic acids and derivatives, benzene and substituted derivatives, and prenol lipids are the most abundant chemical classes observed in LC-MS/MS analysis for the three species of *Suillus* used in this study.

subclasses of terpenoids out of which terpene lactones (13), diterpenoids (8), and sesquiterpenoids (7) were the three most abundant subclasses.

Overall, the majority of these putative terpenes were produced in both monoculture and coculture. A subset of terpenes that were relatively abundant in the data set were further processed using the program Skyline (25, 26) to manually curate metabolite feature extraction for refined quantification. The Skyline results can be found in MassIVE (see Data Availability). The abundance of these distinct terpenes varied greatly among the culture conditions (Fig. 4A). For example, the metabolite feature matching to the non-volatile diterpenoid sandaracopimaric acid was relatively more abundant in coculture conditions, while the monoterpenoid, myrtenal, was relatively more abundant in the monoculture of *S. hirtellus* EM16 than when *S. hirtellus* EM16 was cocultured with the other two species. Interestingly, sandaracopimaric acid has been previously quantified in pine resin (32) and seedling tissue where it was found to play an important ecological role in interactions between plant and other community members. In an effort to further validate the identification of sandaracopimaric acid in this study, a commercial standard was purchased (A2B Chem, CAS No.: 471–74-9) and measured using the same LC-MS/MS settings. As shown in Fig. 4B, there is a high similarity between the tandem mass spectra for the sandaracopimaric acid analytical standard and the *Suillus*-derived metabolite. Previous work studying pine resin and plant tissue has shown that these chemical fractions also have high concentrations of other non-volatile diterpenoids, including abietic acid, pimaric acid, and isopimaric acid (33–35). To assess tandem mass spectra fragmentation similarity between the isomers sandaracopimaric and abietic acid, we purchased a commercial standard for abietic acid (Sigma Aldrich, CAS No.: 514–10-3), measured using the same LC-MS/MS settings. As shown in Fig. 4C, the spectral matching between the *Suillus*-derived metabolite and this abietic acid analytical standard was also quite similar. Based on this piece of data alone, it is equally likely that this metabolite is either sandaracopimaric acid or abietic acid.

## Classical molecular networking illustrates *Suillus* exometabolite diversity and distinct chemical profiles

Molecular networking was used to visualize and explore the entire chemical space beyond what was uncovered by the Compound Discoverer workflow. All experimental

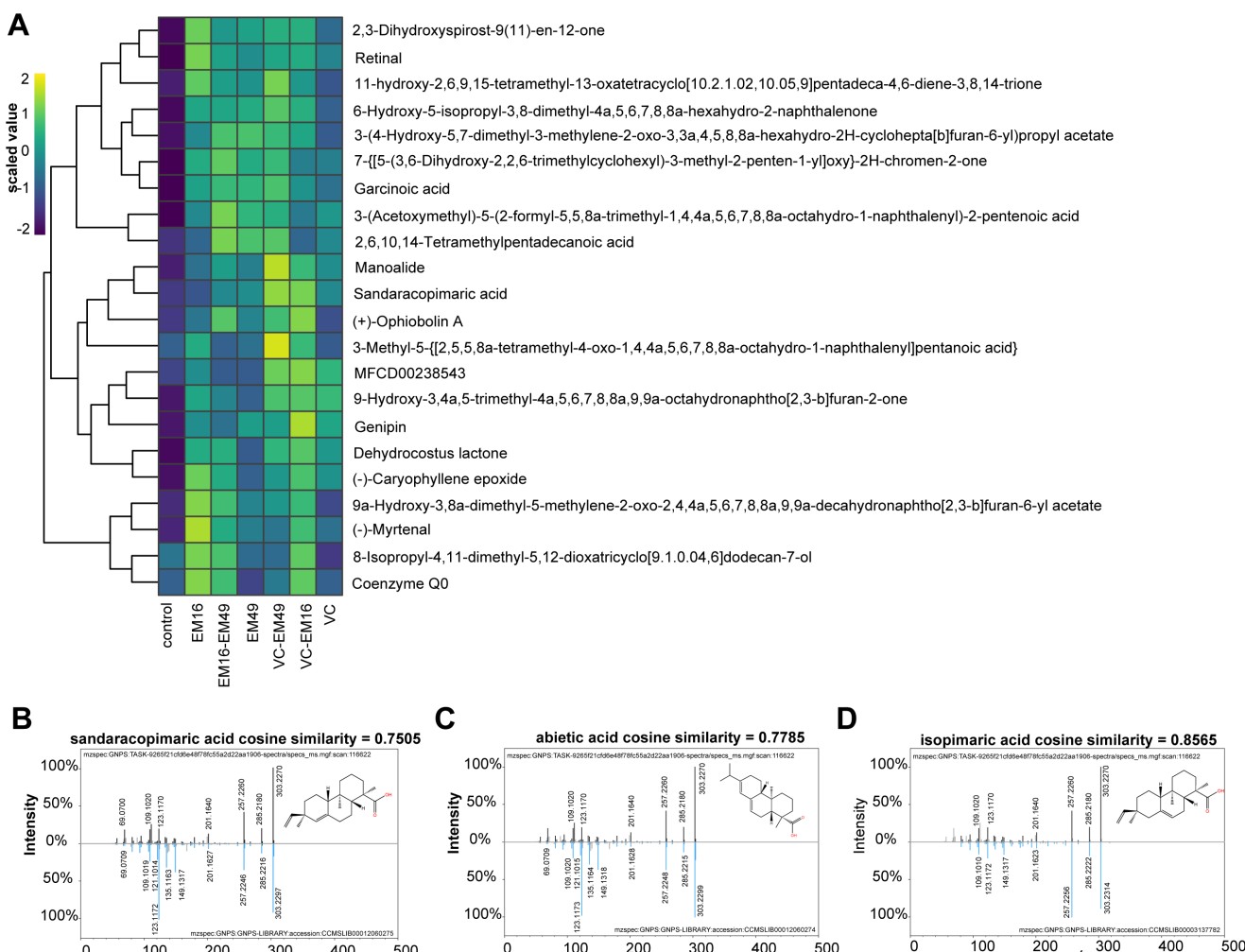

**FIG 4** Prenol lipids relative abundance varied among culture conditions. (A) A heatmap illustrates the relative abundances of selected prenol lipids metabolites that varied in their abundance among culture conditions in ethylacetate fraction. These putative identifications were the result of an untargeted metabolomics analysis using Compound Discoverer. For each compound, the average intensity across biological replicates was Log10 transformed into a scaled value. Mirror match images for (B) sandaracopimaric acid, (C) abietic acid, and (D) isopimaric acid. The experimentally observed MS/MS spectrum is shown at the top, and a representative MS/MS spectrum from a pure standard is shown in blue color at the bottom.

data were processed on the GNPS analysis platform (https://gnps.ucsd.edu/; accessed 28 June 2023) in a single workflow using default settings to create a classical molecular network that connects experimental MS/MS spectra (nodes) using a cosine scoring scheme (edges) ranging from 0 (totally dissimilar) to 1 (completely identical). The raw .mzML files from the study were separated into five different groups when performing classical molecular networking. All the monoculture files were grouped as one, all three coculture combinations were separately grouped, and all media-only controls were grouped together, thus resulting in five different groups. Experimental MS/MS spectra were matched against the GNPS-community spectral library (a relatively large collection of publicly accessible natural product and metabolomics MS/MS data) to assign putative annotations and identify molecular families, which are defined as related MS/MS spectra differing by simple structural or chemical transformations. The MolNetEnhancer workflow (36) in GNPS was used to combine the outputs from molecular networking and the automated chemical classification through ClassyFire to provide a more comprehensive chemical overview (Table S5). With the applied settings, the molecular network contained 16,904 nodes with ~14% associated with a chemical classification (Fig. S4; Fig. 5). Among

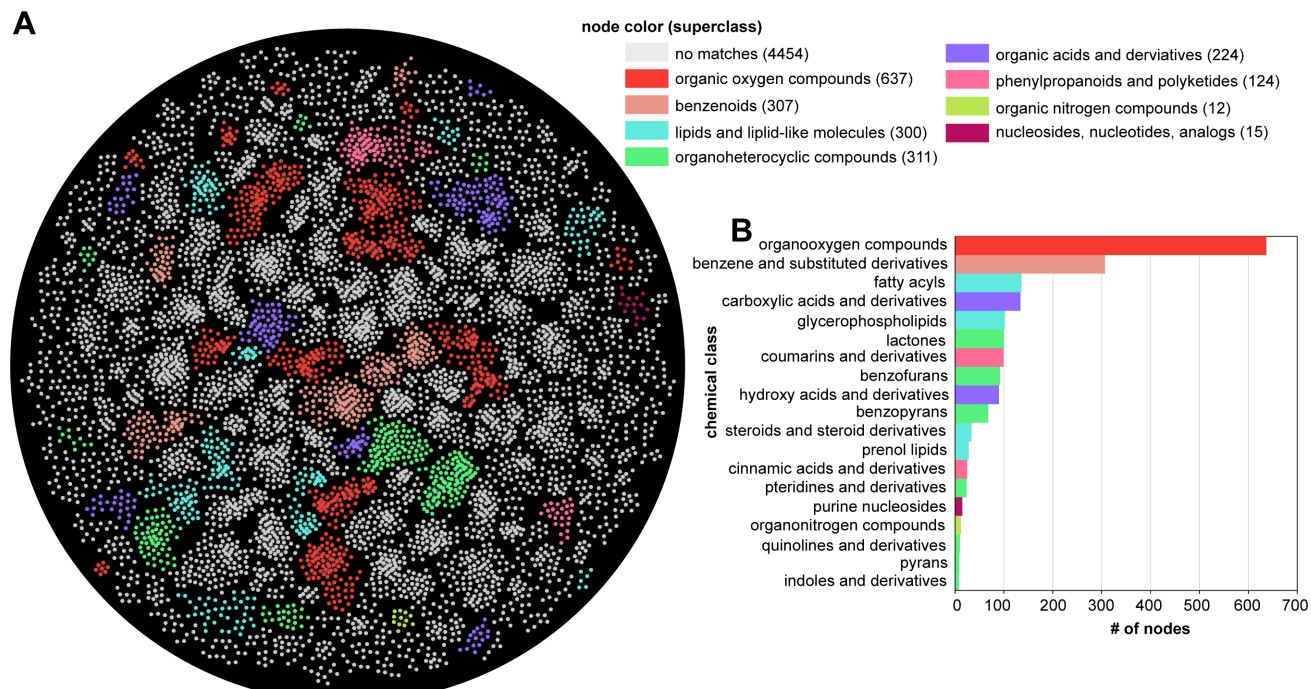

**FIG 5** Classical molecular network uncovered predominant annotated and unannotated MS/MS spectra. (A) The GNPS-derived molecular network was visualized by the Cytoscape software. Each node represents an MS/MS spectrum from this study. Nodes colored white represent unannotated MS/MS spectrum, and colored nodes represent an MS/MS spectrum associated with a putative metabolite annotation with a relatively high spectral similarity score (correlation >0.7). Nodes with putative annotations were annotated by the MolNetEnhancer workflow in GNPS to illustrate the different superclass annotations. Only a subset of the molecular network is shown by removing subnetworks with fewer than six nodes. (B) A bar plot shows the number of metabolites belonging to a chemical class reported by the MolNetEnhancer workflow in GNPS. The full molecular network and associated chemical classes are illustrated in Fig. S4.

all the nodes, ~68% of the nodes belonged to subnetworks with at least two MS/MS spectra. Analysis of all 16,904 nodes showed that a total of 2,925 nodes were unique to monoculture conditions, 2,482 nodes were unique to coculture conditions, and 1,780 nodes were observed in all the culture conditions yet absent in media blanks. Across the entire study, GNPS provided putative annotations for 56 metabolites, which were representative of nine chemical superclasses. In general, the top 20 chemical classes and their representation according to GNPS were similar to what was found using the Compound Discoverer workflow (Fig. S4). Further inspection of the resulting prenol lipid annotations revealed that the sandaracopimaric acid-related MS/MS spectra (Fig. 4B) from the Compound Discoverer workflow matched with high confidence to the stereoisomer isopamaric acid in the GNPS public library (Fig. 4D).

## DISCUSSION

Pine trees are native to the northern hemisphere but have been introduced globally for shade, shelter, and wood products, making them some of the most ecologically and economically significant tree groups. In boreal forests, which are often nutrient poor, pine nutrition is largely supported by associations with ECM fungi such as *Suillus*. The establishment and maintenance of these ecological interactions depend on a combination of factors, including effective communication between the host and fungal partner. Secondary metabolites are well known for their role in communication between and within species. When secondary metabolites are released into the extracellular environment (exometabolites), they can convey information about an organism's presence, function, or metabolic status. These signals help organisms recognize resources and threats. Typically, fungal secondary metabolites are not constitutively expressed, and

require specific metabolic or environmental cues for induction—complicating the characterization of these important chemical signals.

Previous whole-genome analysis suggested that the *Suillus* species have a relatively large number of terpene and NRPS-like BGCs compared to other ECM fungi (12). To further assess these highly specialized genomes, we used genome mining and orthology analysis to predict and compare BGCs in three *Suillus* species (*S. hirtellus* EM16, *S. decipiens* EM49, and *S. cothurnatus* VC1858). We found that these three species encoded an abundance of BGCs, which were dominated by species-specific clusters that displayed little conservation between species. In agreement with previous predictions made across the genus (12), the majority of these clusters were composed of BGCs containing terpene and NRPS-like backbones.

Next, we employed metabolomics to characterize the extracellular secondary metabolites produced by *Suillus* under coculture and monoculture conditions. Conducting a global assessment of metabolites is a significant challenge because variation in sample preparation and methods for metabolite detection can introduce biases. This motivated us to perform a biphasic extraction to wholistically sample and measure polar and nonpolar fractions of secondary metabolites. It is important to note that no single analytical platform is suitable for all metabolomic studies, and the selection of which platform to use should be guided by the research question and the nature of the metabolites of interest (37, 38). For this study, LC-MS was selected for untargeted metabolomic assessment because this method offers high sensitivity, selectivity, and comprehensive coverage for a diverse set of metabolites with different chemical properties (39, 40). Empowered by the growing availability of LC-MS public data resources, we sought to leverage several available spectral libraries to address another challenge associated with untargeted metabolomics—metabolite identification. In this study, secondary metabolites produced under coculture conditions were mostly fatty acyls, carboxylic acids and derivatives, benzene and substituted derivatives, prenol lipids, and organic oxygen compounds. As expected, only a small amount of the total data (<20%) could be assigned a putative annotation based on spectral matching against a public reference library. While these annotations are useful to describe the physiochemistry of the secondary metabolites observed, it is important to further note that these are putative identifications that require additional verification. The use of commercial standards is recommended to achieve a higher level of confidence. However, as shown for tandem mass spectra matching to several isomers (sandaracopimaric acid, isopimaric acid, and abietic acid), other orthogonal techniques (e.g., nuclear magnetic resonance spectroscopy) must be used to complete the identification process. Nevertheless, the high-mass-accuracy LC-MS putative identifications presented in this study revealed a vast, yet to be characterized, landscape of unknown compounds while providing new insights into the chemical ecology of *Suillus* fungi.

The exometabolome of *Suillus* was found to be rich in terpenes, and this observation aligned well with the derived genome-mining predictions. Given the ecological importance of terpenes for pine growth and defense, we manually curated quantitative abundances for a subset of putative terpene identifications. Among these, we further interrogated tandem mass spectra belonging to the non-volatile diterpenoid sandaracopimaric acid through an assessment against several available analytical standards. Because the spectral matching between stereoisomer and isomer candidates cannot be differentiated, additional experiments are needed to complete the process of identification. Nevertheless, the prospect of *Suillus* fungi producing either of these compounds is intriguing considering that both are typically produced by conifers such as pines (41).

While the ecological role of terpene production in ECM fungi is largely unknown, recent metatranscriptomic analysis of pine roots inoculated with different *Suillus* species revealed that terpene synthase genes were differentially expressed during incompatible-host parings (42) suggesting that they may play a role in recognition and stress responses. The identification of diterpene acids in our *Suillus* cocultures raises several new questions about the ecological roles and origins of terpenes in ECM fungi. Diterpene

acids are known to have diverse functions in fungal community interactions; abietic acid is a well-known elicitor of spore germination in *Suillus* (43, 44) and has been shown to be antifungal against conifer pathogens such as *Heterobasidion* and *Ophiostoma* (45, 46). While diterpene acid production is not unheard of in fungi (47), the production of these compounds is typically associated with plants—particularly conifers, raising questions about the origin of diterpene acid BCGs in *Suillus*. Identifying the exact BGCs responsible for encoding these secondary metabolites is nontrivial, but further work to link these products to genes and determine the origin of diterpene acid production in *Suillus* (including the potential for horizonal gene transfer from the host) is warranted.

In conclusion, genome mining coupled with co-culture and untargeted metabolomics revealed a diverse set of secondary metabolites likely to be important for ECM community interactions. In agreement with previous studies, we identified a large number of terpene BGCs using genome mining, encompassing 62 unique clusters. Importantly, while our coculture BGC induction method proved highly effective in inducing a variety of terpenes, these within-genus pairings represent only a single type of environmental trigger, and the true terpene diversity of *Suillus* is likely even greater than what we have reported here. Taken together, our LC-MS/MS-based untargeted metabolomics analysis of the *Suillus* exometabolome revealed diverse terpenes exuded differentially from species grown in monoculture- and coculture-specific conditions, highlighting the potential role these chemicals may play in inter- and intraspecific community interfacing.

## ACKNOWLEDGMENTS

This research is supported by the Plant–Microbe Interfaces (https://pmiweb.ornl.gov/) Scientific Focus Areas at ORNL and by a subaward DE-SC0020403 to R.V. through the U.S. Department of Energy's Office of Biological and Environmental Research. L.L. is supported by funding from the National Institutes of Health grant no. T32-AI052080 via the Tri-I MMPTP Fellowship. S.M. is supported by funding from the Secure Ecosystem Engineering and Design (https://seed-sfa.ornl.gov/) Scientific Focus Area funded by the Genomic Science Program of the U.S. Department of Energy, Office of Science, Office of Biological and Environmental Research (BER) as part of the Secure Biosystems Design Science Focus Area (SFA).

## AUTHOR AFFILIATIONS

[1]UT-ORNL Graduate School of Genome Science and Technology, University of Tennessee, Knoxville, Tennessee, USA
[2]Biosciences Division, Oak Ridge National Laboratory, Oak Ridge, Tennessee, USA
[3]Biology Department, Duke University, Durham, North Carolina, USA

## AUTHOR ORCIDs

Sameer Mudbhari (iD) http://orcid.org/0000-0002-3091-7544
Paul E. Abraham (iD) http://orcid.org/0000-0003-2685-9123

## FUNDING

| Funder | Grant(s) | Author(s) |
| --- | --- | --- |
| U.S. Department of Energy (DOE) | Plant–Microbe Interfaces Scientific Focus Area | Robert L. Hettich |
| | | Rytas Vilgalys |
| | | Paul E. Abraham |
| | | Lotus Lofgren |
| | | Manasa R. Appidi |
| U.S. Department of Energy (DOE) | Secure Ecosystem Engineering and Design Scientific Focus Area | Paul E. Abraham |
| | | Sameer Mudbhari |

## AUTHOR CONTRIBUTIONS

Sameer Mudbhari, Data curation, Formal analysis, Investigation, Methodology, Validation, Visualization, Writing – original draft | Lotus Lofgren, Conceptualization, Data curation, Formal analysis, Investigation, Methodology, Project administration, Resources, Supervision, Validation, Visualization, Writing – original draft, Writing – review and editing | Manasa R. Appidi, Methodology, Writing – review and editing | Rytas Vilgalys, Funding acquisition, Investigation, Project administration, Resources, Supervision, Writing – review and editing | Robert L. Hettich, Supervision, Writing – review and editing | Paul E. Abraham, Conceptualization, Data curation, Formal analysis, Funding acquisition, Investigation, Methodology, Project administration, Resources, Supervision, Validation, Visualization, Writing – original draft, Writing – review and editing

## DATA AVAILABILITY

The antiSMASH, BiG-SCAPE, and data from LC-ESI-MS/MS analyses used in this study are all publicly online at MassIVE (https:// massive.ucsd.edu/) with the assigned accession MSV000094072. Similarly, the GNPS-based classical molecular network result file can be accessed at https://gnps.ucsd.edu/ProteoSAFe/status.jsp?task=fe2b002ce423463da78e55d8adde536f#.

## ADDITIONAL FILES

The following material is available online.

### Supplemental Material

**Figure S1 (mSystems01225-23-s0001.tif).** Growth differences observed for *Suillus* species grown in intra- and inter-species pairings.
**Figure S2 (mSystems01225-23-s0002.tif).** The chemical diversity for the putatively identified metabolites in the aqueous fraction.
**Figure S3 (mSystems01225-23-s0003.tif).** The chemical diversity for the putatively identified metabolites in the organic fraction.
**Figure S4 (mSystems01225-23-s0004.tif).** Classical molecular network uncovered predominant annotated and unannotated MS/MS spectra.
**Supplemental File (mSystems01225-23-s0005.docx).** Additional details provided for LC-MS methods and supplemental legends.
**Table S1 (mSystems01225-23-s0006.xlsx).** Verbose data matrix of identified compound features generated from the software Compound Discoverer v3.3.
**Table S2 (mSystems01225-23-s0007.xlsx).** Subset of Compound Discoverer verbose data matrix filtered to compound features that matched with confidence against mzCloud and/or NIST2020 spectral libraries.
**Table S3 (mSystems01225-23-s0008.xlsx).** A non-redundant list of spectral matched compound features filtered to those with abundances significantly greater than media blank.
**Table S4 (mSystems01225-23-s0009.xlsx).** ClassyFire chemical classifications are listed for metabolites identified from the Compound Discoverer workflow.
**Table S5 (mSystems01225-23-s0010.xlsx).** A data matrix of putative metabolites identified by the GNPS classical molecular network workflow.

### Open Peer Review

**PEER REVIEW HISTORY (review-history.pdf).** An accounting of the reviewer comments and feedback.

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
