## [Reviewer comments · mSystems]

Decoding the chemical language of *Suillus* fungi: genome mining and untargeted metabolomics uncover terpene chemical diversity

Sameer Mudbhari, Lotus Lofgren, Manasa Appidi, Rytas Vilgalys, Robert Hettich, and Paul Abraham

Corresponding Author(s): Paul Abraham, Oak Ridge National Laboratory

Review Timeline:

Submission Date:	November 17, 2023
Editorial Decision:	December 15, 2023
Revision Received:	February 14, 2024
Editorial Decision:	February 15, 2024
Revision Received:	February 15, 2024
Accepted:	February 19, 2024

Editor: Yu-Liang Yang

Reviewer(s): Disclosure of reviewer identity is with reference to reviewer comments included in decision letter(s). The following individuals involved in review of your submission have agreed to reveal their identity: Kristina Haslinger (Reviewer #1); Emile Gluck-Thaler (Reviewer #2)

Transaction Report:

DOI: <https://doi.org/10.1128/msystems.01225-23>

Re: mSystems01225-23 (Decoding the chemical language of *Suillus* fungi: genome mining and untargeted metabolomics uncover terpene chemical diversity)

Dear Dr. Paul E. Abraham:

Revision Guidelines

Sincerely,
Yu-Liang Yang
Editor
mSystems

Reviewer #1 (Comments for the Author):

Please find the attachment.

Reviewer #2 (Comments for the Author):

In this study, Mudbhari et al conduct an in-depth investigation into the exometabolomes of three *Suillus* species, both in pure culture and co-culture. As expected, they find that many new metabolites are produced when different isolates are grown in co-culture with each other. The authors do a great job summarizing a lot of metabolomic data, and I commend them on a clear and

succinct manuscript. I have just a few comments and suggestions that I hope would improve the manuscript.

180-203: I know similar antiSMASH analyses have previously been published, but since new analyses are being reported here, please provide all data associated with the BGC predictions in these genomes as supplementary info (e.g., GFF files of all predicted clusters and their genomic coordinates, antiSMASH HTML files, all bigSCAPE files showing GCF assignments for each BGC etc etc).

Figure 5: Including all "no matches" nodes makes it difficult to appreciate and visualize the network structure. I suggest moving panel A to the supplement doing one of two things for the main figure to simplify the network for visualization purposes: either remove all networks containing fewer than 10 nodes (or some other reasonable threshold) or remove all networks consisting entirely of nodes with "no matches".

Given all the incredible data presented here, I think a couple of additional statistical analyses that might increase interest in this study are justified:

Figure 4A: the heatmap makes it difficult to appreciate whether changes in relative abundance are due to additive or interactive effects between co-cultured fungi but this would seem important to determine given the objectives of this study. e.g., does methylpentanoic acid have a higher abundance in VC-EM16 co-culture simply because its production in VC and its production in EM16 are being added together or is there more of this compound than you would expect? Please explore statistical tools like a generalized linear model or consult with a statistician about another appropriate method where you can estimate additive and non-additive variance in terpene abundance as a function of species 1, species 2, and species 1 x species 2

Figure 3: Its very useful to see in some of the other figures how metabolites break down according to treatment (e.g., figure 2c). Given the focus on suillus chemical ecology, I think a similar breakdown would be interesting in this figure where you specifically examine different chemical classes. For example, how many chemicals in each class are only produced in monoculture vs co-culture? As a complement to this bar graph, is there a way to calculate which combinations of fungi produce the greatest diversity of metabolite chemical classes? Is it possible to calculate and compare alpha and beta "chemical" diversity and compare them between treatments? Does this in any way correlate with the number of BGCs in the interacting genomes?

Mudbhari et al. performed untargeted metabolomics of three *Suillus* species, an important genus of ectomycorrhizal fungi. The study describes the mass spectrometric analysis of metabolites produced by the three species grown individually and in co-culture. The authors attempted to identify and/or classify the metabolites with two different approaches that are both well-established in the field.

The structure and language of this manuscript is clear and easy to follow. I appreciate that the authors are very cautious in interpreting their results, since they did not identify the metabolites by NMR. However, this makes the study seem rather incomplete and shallow. The authors did not even speculate on the biosynthetic origin of the metabolites although they point out that there are multiple biosynthetic gene clusters in the genomes of these species. The BGC analysis, is a repetition of a previous analysis when the genomes were first published.

In addition to this lack of novelty/innovation, my major concern is that the methods section is very brief and does not properly explain how the samples were prepared (replicates, controls, etc.) and how the analysis of the metabolomics data was performed (blanks, QC, etc.).

Minor concerns:

Line 117: what exactly was the input for BiG-SCAPE?

Line 122: where did the specimen come from? Are the strains identical with the ones sequenced in ref 12?

Line 125: Was each experiment performed with 5 petri dishes and then analyzed in 3 replicates, so 15 replicates in total? I find it confusing that the 5 replicates are called technical, since I would consider those biological and the other 3 technical replicates.

Line 169: I would expect a separate methods section explaining the data processing (QC, background subtraction, etc.) and analysis, as well as the statistical analysis.

Line 193: Why are the numbers of NRPS-like and terpene BGCs in panel C and D different from panel B?

Line 211: Unclear description of the steps of data analysis and the rationale behind it. The procedure should be properly explained in the methods section. What was the rationale for analyzing the data with ChemSpider and then excluding it afterwards?

Line 215: Why was PCA not directly performed after the spectral matching using Compound Discoverer even using datasets without any analysis steps, which would be the most original datasets to check the quality of the data? Alternatively, why not perform PCA on the final list of 1,118 metabolites? The entire analysis appears rather arbitrary without further explanation of the rationale.

Line 215: Please use "biological" and "technical" replicates more consistently (see comment

above).

Line 274: Do all reference compounds also elute with the same retention time?

Line 303: typo "4D"?

Line 423: Please define the number and nature of replicates here.

Line 427: panel labels A/B/C should be B/C/D.

Figure 2A/B: if there were 15 replicates, why are there only 3-5 dots per group in this plot?

Figure 2C: VC has no unique metabolites? Why is there no black line in the third last column?

In Figure 4A: what exactly does the scaled value mean? How was it calculated? Without that information it is impossible to interpret the heatmap and see which metabolites are up- or downregulated between co-culture and monoculture.

Reply to reviewer's comments:

Reviewer 1 Mudbhari et al. performed untargeted metabolomics of three *Suillus* species, an important genus of ectomycorrhizal fungi. The study describes the mass spectrometric analysis of metabolites produced by the three species grown individually and in co-culture. The authors attempted to identify and/or classify the metabolites with two different approaches that are both well-established in the field. The structure and language of this manuscript is clear and easy to follow. I appreciate that the authors are very cautious in interpreting their results, since they did not identify the metabolites by NMR. However, this makes the study seem rather incomplete and shallow. The authors did not even speculate on the biosynthetic origin of the metabolites although they point out that there are multiple biosynthetic gene clusters in the genomes of these species.

Response to reviewer: We thank the reviewer for their critical evaluation of our work and the thoughtful remarks and constructive criticisms. We understand that there are additional measurements and experiments that can be performed to define the structure of the compounds and their relatedness to the predicted biosynthetic gene clusters. We agree with the reviewer that those pursuits are warranted. Although we do intend to further characterize many of the resulting compounds observed, those experiments (e.g., NMR analysis and genome engineering) do require a fair amount of additional time, resources, and energy that fall outside the scope of this study. At this point in time, we prefer to not speculate as to what metabolites match against any particular antiSMASH prediction. This is largely because predicted backbone genes, such as terpene synthase, could lead to the production of a variety of terpene molecules.

Reviewer 1 The BGC analysis, is a repetition of a previous analysis when the genomes were first published.

Response to reviewer: While we did perform BGC analysis in our previously published research article (Lofgren et al., *New Phytologist*, 2021) and in this study, the analysis and outcome are distinct. In the already published study, our comparative genomics analysis investigated BGC predictions and their similarities across many genera of ECM fungi. In this study, we instead investigated BGC predictions and their similarities across three distinct *Suillus* species. Here, this study revealed that there is inter-species variability in the encoded biosynthetic gene clusters associated with secondary metabolite production. To further clarify this point, we have revised the manuscript text on page 6, line 231-254.

Reviewer 1 In addition to this lack of novelty/innovation, my major concern is that the methods section is very brief and does not properly explain how the samples were prepared (replicates, controls, etc.) and how the analysis of the metabolomics data was performed (blanks, QC, etc.).

Response to reviewer: We thank the reviewer for their suggestion to further develop the methods section and include additional information. As suggested, we have revised the manuscript text on page 5, line 173-220.

Reviewer 1 I appreciate that the authors are very cautious in interpreting their results, since they did not identify the metabolites by NMR. However, this makes the study seem rather incomplete and shallow.

Author's comment: We thank the reviewer for their assessment and their encouragement to pursue additional levels of characterization to further the impact of these findings. As mentioned in a previous comment, these efforts are quite substantial and require a separate publication.

Reviewer 1 Line 117: what exactly was the input for BiG-SCAPE?

Response to reviewer: The input for BiG-SCAPE is the GenBank files obtained from antiSMASH. We have revised the text to include this information, and this can be found on page 3, line 119-120.

Reviewer 1 Line 122: where did the specimen come from?

Response to reviewer: The cultures originally came from fruitbodies growing under *Pinus* species. We have updated the manuscript with this information in page 3, line 122-123.

Reviewer 1 Line 125: Was each experiment performed with 5 petri dishes and then analyzed in 3 replicates, so 15 replicates in total? I find it confusing that the 5 replicates are called technical, since I would consider those biological and the other 3 technical replicates.

Response to reviewer: We used five replicates of each species and we considered these to be biological replicates. These 5 replicates were then used for each condition. We have revised the manuscript to make it clear to understand and these changes can be found on page 4, line 128.

Reviewer 1 Line 169: I would expect a separate methods section explaining the data processing (QC, background subtraction, etc.) and analysis, as well as the statistical analysis.

Response to reviewer: As suggested, we added a separate section to describe our methods that includes text associated to data processing, analysis, and statistical testing. This new section can be found on page 5-6, line 168-216.

Reviewer 1 Line 193: Why are the numbers of NRPS-like and terpene BGCs in panel C and D different from panel B? Unclear description of the steps of data analysis and the rationale behind it. The procedure should be properly explained in the methods section

Response to reviewer: The differences in counts between Figure 1 panels B-D is because UpSet plots are always based on binary data representations, so they condense instances where there are multiple copies of the same cluster into single unique observations. There are not major differences in copy number variation for metabolite clusters in this set, but there were a couple of duplications, and those being condensed in these plots contributes to observed count differences between the graphs.

Reviewer 1 Line 211: What was the rationale for analyzing the data with ChemSpider and then excluding it afterwards?

Response to reviewer: We thank the reviewer for noting the confusion in the current version of the manuscript. To address, the untargeted workflow that we used in Compound discoverer provides peak area for all features that have a measured m/z and retention time. However, not all of these features could be confidently assigned to a particular metabolite name because of only limited tandem MS data in the standard library such as NIST2020 and mzCloud. The ChemSpider was initially used to make it easy for us to follow up on certain features even if it does not have tandem MS data. To make our identification based on tandem MS data, we only did further analysis on those features that matched with NIST2020 and mzCloud library.

Reviewer 1 Line 215: Why was PCA not directly performed after the spectral matching using Compound Discoverer even using datasets without any analysis steps, which would be the most original datasets to check the quality of the data? Alternatively, why not perform PCA on the final list of 1,118 metabolites?

The entire analysis appears rather arbitrary without further explanation of the rationale. Line 215: Please use “biological” and “technical” replicates more consistently (see comment above).

Response to reviewer: We subset the data for PCA to only focus on those metabolites have tandem MS data and a putative identification to our spectral libraries. This decision to only focus on putative metabolite identifications was to be consistent and more accurate with our other downstream analyses. We understand that this rationale was not clear because we did not properly explain how we were addressing “redundancies” in the data. To address this, we have added additional text to explain our rationale for why the number of putative metabolite identifications was reduced from 3,769, and this can be found on page 7, line 267. Additionally, we addressed confusion related to the usage of “biological” and “technical” replicates in a previous comment.

Response to reviewer:

Reviewer 1 Line 274: Do all reference compounds also elute with the same retention time? Line 303: typo “4D”?

Response to reviewer: That is a great question and could help with classification. Unfortunately, compounds had similar elution times.

Reviewer 1 Line 303: typo “4D”?

Response to reviewer: We corrected this typo on page 9, line 356.

Reviewer 1 Line 423: Please define the number and nature of replicates here.

Response to reviewer: As mentioned in a previous comment to the reviewer, we used five replicates of each culture conditions. We have revised the manuscript to make it clear to understand and these changes can be found on page 125-127.

Reviewer 1 Line 427: panel labels A/B/C should be B/C/D.

Response to reviewer: As suggested, we have revised the text page 12, line 482.

Reviewer 1 Figure 2A/B: if there were 15 replicates, why are there only 3-5 dots per group in this plot?

Response to reviewer: We have 3 replicates (i.e., dots) for the control (media only) and 5 biological replicates (dots) per sample group. Note, we lost a biological replicate for one sample group.

Reviewer 1 Figure 2C: VC has no unique metabolites? Why is there no black line in the third last column?

Response to reviewer: We saw putatively identified metabolites that unique to VC in aqueous extract (Figure 2D). However, no unique VC specific metabolite was seen in organic extract (Figure 2C).

Reviewer 1 In Figure 4A: what exactly does the scaled value mean? How was it calculated? Without that information it is impossible to interpret the heatmap and see which metabolites are up- or downregulated between co-culture and monoculture.

Response to reviewer: For each compound, the average intensity across biological replicates was Log10-transformed into a scaled value (standard z-score). This data transformation was performed to accommodate the large range of values measured. We have revised the manuscript by adding this information to the figure legend on page 12, line 280-281.

Reviewer #2 (Comments for the Author):

Reviewer 2 In this study, Mudbhari et al conduct an in-depth investigation into the exometabolomes of three *Suillus* species, both in pure culture and co-culture. As expected, they find that many new metabolites are produced when different isolates are grown in co-culture with each other. The authors do a great job summarizing a lot of metabolomic data, and I commend them on a clear and succinct manuscript. I have just a few comments and suggestions that I hope would improve the manuscript.

Response to reviewer: We thank the reviewer for their thoughtful remarks and constructive criticism.

Reviewer 2 180-203: I know similar antiSMASH analyses have previously been published, but since new analyses are being reported here, please provide all data associated with the BGC predictions in these genomes as supplementary info (e.g., GFF files of all predicted clusters and their genomic coordinates, antiSMASH HTML files, all bigSCAPE files showing GCF assignments for each BGC etc etc).

Response to reviewer: We have provided the requested information as new supplementary data files detailed in the 'Data Availability' statement. Further, this information has been provided in revised manuscript on page 5, line 249-250.

Reviewer 2 Figure 5: Including all "no matches" nodes makes it difficult to appreciate and visualize the network structure. I suggest moving panel A to the supplement doing one of two things for the main figure to simplify the network for visualization purposes: either remove all networks containing fewer than 10 nodes (or some other reasonable threshold) or remove all networks consisting entirely of nodes with "no matches".

Response to reviewer: We thank the reviewer for the feedback on how to best present these results. We have revised Figure 5, as suggested, by moving this version to Supplemental Material (Figure S4). We have replaced the original with a revised version that filtered out all subnetworks containing fewer than 6 nodes. We decided to keep the nodes with "no matches" because we view these as an important feature of the data generated.

Reviewer 2 Figure 4A: the heatmap makes it difficult to appreciate whether changes in relative abundance are due to additive or interactive effects between co-cultured fungi but this would seem important to determine given the objectives of this study. e.g., does methylpentanoic acid have a higher abundance in VC-EM16 co-culture simply because its production in VC and its production in EM16 are being added together or is there more of this compound than you would expect? Please explore statistical tools like a generalized linear model or consult with a statistician about another appropriate method where you can estimate additive and non-additive variance in terpene abundance as a function of species 1, species 2, and species 1 x species 2

Author's reply: We agree with the reviewer that it would be helpful to dissect the amount of compound signal being provided by each organism present in the coculture. That is, what amount of increased signal in coculture could be explained by either a combined signal sourced from both organisms (additive) or by enhanced gene expression of a single organism (interactive). We have carefully considered this comment; however, our expectation is that those determinations (i.e., additive effects) are best made through alternative approaches, such as stable isotope labeling, and additional experimentation.

Reviewer 2 Figure 3: Its very useful to see in some of the other figures how metabolites break down according to treatment (e.g., figure 2c). Given the focus on suillus chemical ecology, I think a similar breakdown would be interesting in this figure where you specifically examine different chemical classes. For example, how many chemicals in each class are only produced in monoculture vs co-culture? As a complement to this bar graph, is there a way to calculate which combinations of fungi produce the greatest diversity of metabolite chemical classes? Is it possible to calculate and compare alpha and beta "chemical" diversity and compare them between treatments? Does this in any way correlate with the number of BGCs in the interacting genomes?

Author's reply: We thank the reviewer for the thoughtful comments. We have created new figures (Figures S2 and S3) to address these comments for the organic and aqueous fractions. These additional UpSet plots illustrate the overlap in chemical classes for the observed metabolites in each sample.

Re: mSystems01225-23R1 (Decoding the chemical language of *Suillus* fungi: genome mining and untargeted metabolomics uncover terpene chemical diversity)

Dear Dr. Paul E. Abraham:

Thank you for the privilege of reviewing your work. The manuscript has been revised accordingly and addresses the feedback provided by the reviewers. While you have made significant progress, there are several minor issues that still require attention prior to formal acceptance. Kindly refer to the annotations in the attached document for further details.

Please return the manuscript as soon as possible; if you cannot complete the modification within this time period, please contact me. If you do not wish to modify the manuscript and prefer to submit it to another journal, notify me immediately so that the manuscript may be formally withdrawn from consideration by mSystems.

Revision Guidelines

Sincerely,
Yu-Liang Yang
Editor
mSystems

Reply to editor:

As requested, we have addressed the highlighted marks in the manuscript.

Reply to reviewer's comments:

Reviewer 1 Mudbhari et al. performed untargeted metabolomics of three *Suillus* species, an important genus of ectomycorrhizal fungi. The study describes the mass spectrometric analysis of metabolites produced by the three species grown individually and in co-culture. The authors attempted to identify and/or classify the metabolites with two different approaches that are both well-established in the field. The structure and language of this manuscript is clear and easy to follow. I appreciate that the authors are very cautious in interpreting their results, since they did not identify the metabolites by NMR. However, this makes the study seem rather incomplete and shallow. The authors did not even speculate on the biosynthetic origin of the metabolites although they point out that there are multiple biosynthetic gene clusters in the genomes of these species.

Response to reviewer: We thank the reviewer for their critical evaluation of our work and the thoughtful remarks and constructive criticisms. We understand that there are additional measurements and experiments that can be performed to define the structure of the compounds and their relatedness to the predicted biosynthetic gene clusters. We agree with the reviewer that those pursuits are warranted. Although we do intend to further characterize many of the resulting compounds observed, those experiments (e.g., NMR analysis and genome engineering) do require a fair amount of additional time, resources, and energy that fall outside the scope of this study. At this point in time, we prefer to not speculate as to what metabolites match against any particular antiSMASH prediction. This is largely because predicted backbone genes, such as terpene synthase, could lead to the production of a variety of terpene molecules.

Reviewer 1 The BGC analysis, is a repetition of a previous analysis when the genomes were first published.

Response to reviewer: While we did perform BGC analysis in our previously published research article (Lofgren et al., *New Phytologist*, 2021) and in this study, the analysis and outcome are distinct. In the already published study, our comparative genomics analysis investigated BGC predictions and their similarities across many genera of ECM fungi. In this study, we instead investigated BGC predictions and their similarities across three distinct *Suillus* species. Here, this study revealed that there is inter-species variability in the encoded biosynthetic gene clusters associated with secondary metabolite production. To further clarify this point, we have revised the manuscript text on page 6, line 231-254.

Reviewer 1 In addition to this lack of novelty/innovation, my major concern is that the methods section is very brief and does not properly explain how the samples were prepared (replicates, controls, etc.) and how the analysis of the metabolomics data was performed (blanks, QC, etc.).

Response to reviewer: We thank the reviewer for their suggestion to further develop the methods section and include additional information. As suggested, we have revised the manuscript text on page 5, line 173-220.

Reviewer 1 I appreciate that the authors are very cautious in interpreting their results, since they did not identify the metabolites by NMR. However, this makes the study seem rather incomplete and shallow.

Author's comment: We thank the reviewer for their assessment and their encouragement to pursue additional levels of characterization to further the impact of these findings. As mentioned in a previous comment, these efforts are quite substantial and require a separate publication.

Reviewer 1 Line 117: what exactly was the input for BiG-SCAPE?

Response to reviewer: The input for BiG-SCAPE is the GenBank files obtained from antiSMASH. We have revised the text to include this information, and this can be found on page 3, line 119-120.

Reviewer 1 Line 122: where did the specimen come from?

Response to reviewer: The cultures originally came from fruitbodies growing under *Pinus* species. We have updated the manuscript with this information in page 3, line 122-123.

Reviewer 1 Line 125: Was each experiment performed with 5 petri dishes and then analyzed in 3 replicates, so 15 replicates in total? I find it confusing that the 5 replicates are called technical, since I would consider those biological and the other 3 technical replicates.

Response to reviewer: We used five replicates of each species and we considered these to be biological replicates. These 5 replicates were then used for each condition. We have revised the manuscript to make it clear to understand and these changes can be found on page 4, line 128.

Reviewer 1 Line 169: I would expect a separate methods section explaining the data processing (QC, background subtraction, etc.) and analysis, as well as the statistical analysis.

Response to reviewer: As suggested, we added a separate section to describe our methods that includes text associated to data processing, analysis, and statistical testing. This new section can be found on page 5-6, line 168-216.

Reviewer 1 Line 193: Why are the numbers of NRPS-like and terpene BGCs in panel C and D different from panel B? Unclear description of the steps of data analysis and the rationale behind it. The procedure should be properly explained in the methods section

Response to reviewer: The differences in counts between Figure 1 panels B-D is because UpSet plots are always based on binary data representations, so they condense instances where there are multiple copies of the same cluster into single unique observations. There are not major differences in copy number variation for metabolite clusters in this set, but there were a couple of duplications, and those being condensed in these plots contributes to observed count differences between the graphs.

Reviewer 1 Line 211: What was the rationale for analyzing the data with ChemSpider and then excluding it afterwards?

Response to reviewer: We thank the reviewer for noting the confusion in the current version of the manuscript. To address, the untargeted workflow that we used in Compound discoverer provides peak area for all features that have a measured m/z and retention time. However, not all of these features could be confidently assigned to a particular metabolite name because of only limited tandem MS data in the standard library such as NIST2020 and mzCloud. The ChemSpider was initially used to make it easy

for us to follow up on certain features even if it does not have tandem MS data. To make our identification based on tandem MS data, we only did further analysis on those features that matched with NIST2020 and mzCloud library.

Reviewer 1 Line 215: Why was PCA not directly performed after the spectral matching using Compound Discoverer even using datasets without any analysis steps, which would be the most original datasets to check the quality of the data? Alternatively, why not perform PCA on the final list of 1,118 metabolites? The entire analysis appears rather arbitrary without further explanation of the rationale. Line 215: Please use “biological” and “technical” replicates more consistently (see comment above).

Response to reviewer: We subset the data for PCA to only focus on those metabolites have tandem MS data and a putative identification to our spectral libraries. This decision to only focus on putative metabolite identifications was to be consistent and more accurate with our other downstream analyses. We understand that this rationale was not clear because we did not properly explain how we were addressing “redundancies” in the data. To address this, we have added additional text to explain our rationale for why the number of putative metabolite identifications was reduced from 3,769, and this can be found on page 7, line 267. Additionally, we addressed confusion related to the usage of “biological” and “technical” replicates in a previous comment.

Response to reviewer:

Reviewer 1 Line 274: Do all reference compounds also elute with the same retention time? Line 303: typo “4D”?

Response to reviewer: That is a great question and could help with classification. Unfortunately, compounds had similar elution times.

Reviewer 1 Line 303: typo “4D”?

Response to reviewer: We corrected this typo on page 9, line 356.

Reviewer 1 Line 423: Please define the number and nature of replicates here.

Response to reviewer: As mentioned in a previous comment to the reviewer, we used five replicates of each culture conditions. We have revised the manuscript to make it clear to understand and these changes can be found on page 125-127.

Reviewer 1 Line 427: panel labels A/B/C should be B/C/D.

Response to reviewer: As suggested, we have revised the text page 12, line 482.

Reviewer 1 Figure 2A/B: if there were 15 replicates, why are there only 3-5 dots per group in this plot?

Response to reviewer: We have 3 replicates (i.e., dots) for the control (media only) and 5 biological replicates (dots) per sample group. Note, we lost a biological replicate for one sample group.

Reviewer 1 Figure 2C: VC has no unique metabolites? Why is there no black line in the third last column?

Response to reviewer: We saw putatively identified metabolites that unique to VC in aqueous extract (Figure 2D). However, no unique VC specific metabolite was seen in organic extract (Figure 2C).

Reviewer 1 In Figure 4A: what exactly does the scaled value mean? How was it calculated? Without that information it is impossible to interpret the heatmap and see which metabolites are up- or downregulated between co-culture and monoculture.

Response to reviewer: For each compound, the average intensity across biological replicates was Log10-transformed into a scaled value (standard z-score). This data transformation was performed to accommodate the large range of values measured. We have revised the manuscript by adding this information to the figure legend on page 12, line 280-281.

Reviewer #2 (Comments for the Author):

Reviewer 2 In this study, Mudbhari et al conduct an in-depth investigation into the exometabolomes of three *Suillus* species, both in pure culture and co-culture. As expected, they find that many new metabolites are produced when different isolates are grown in co-culture with each other. The authors do a great job summarizing a lot of metabolomic data, and I commend them on a clear and succinct manuscript. I have just a few comments and suggestions that I hope would improve the manuscript.

Response to reviewer: We thank the reviewer for their thoughtful remarks and constructive criticism.

Reviewer 2 180-203: I know similar antiSMASH analyses have previously been published, but since new analyses are being reported here, please provide all data associated with the BGC predictions in these genomes as supplementary info (e.g., GFF files of all predicted clusters and their genomic coordinates, antiSMASH HTML files, all bigSCAPE files showing GCF assignments for each BGC etc etc).

Response to reviewer: We have provided the requested information as new supplementary data files detailed in the 'Data Availability' statement. Further, this information has been provided in revised manuscript on page 5, line 249-250.

Reviewer 2 Figure 5: Including all "no matches" nodes makes it difficult to appreciate and visualize the network structure. I suggest moving panel A to the supplement doing one of two things for the main figure to simplify the network for visualization purposes: either remove all networks containing fewer than 10 nodes (or some other reasonable threshold) or remove all networks consisting entirely of nodes with "no matches".

Response to reviewer: We thank the reviewer for the feedback on how to best present these results. We have revised Figure 5, as suggested, by moving this version to Supplemental Material (Figure S4). We have replaced the original with a revised version that filtered out all subnetworks containing fewer than 6 nodes. We decided to keep the nodes with "no matches" because we view these as an important feature of the data generated.

Reviewer 2 Figure 4A: the heatmap makes it difficult to appreciate whether changes in relative abundance are due to additive or interactive effects between co-cultured fungi but this would seem important to determine given the objectives of this study. e.g., does methylpentanoic acid have a higher abundance in VC-EM16 co-culture simply because its production in VC and its production in EM16 are being added together or is there more of this compound than you would expect? Please explore statistical tools like a generalized linear model or consult with a statistician about another appropriate method where you can estimate additive and non-additive variance in terpene abundance as a function of species 1, species 2, and species 1 x species 2

Author's reply: We agree with the reviewer that it would be helpful to dissect the amount of compound signal being provided by each organism present in the coculture. That is, what amount of increased signal in coculture could be explained by either a combined signal sourced from both organisms (additive) or by enhanced gene expression of a single organism (interactive). We have carefully considered this comment; however, our expectation is that those determinations (i.e., additive effects) are best made through alternative approaches, such as stable isotope labeling, and additional experimentation.

Reviewer 2 Figure 3: It's very useful to see in some of the other figures how metabolites break down according to treatment (e.g., figure 2c). Given the focus on soil chemical ecology, I think a similar breakdown would be interesting in this figure where you specifically examine different chemical classes. For example, how many chemicals in each class are only produced in monoculture vs co-culture? As a complement to this bar graph, is there a way to calculate which combinations of fungi produce the greatest diversity of metabolite chemical classes? Is it possible to calculate and compare alpha and beta "chemical" diversity and compare them between treatments? Does this in any way correlate with the number of BGCs in the interacting genomes?

Author's reply: We thank the reviewer for the thoughtful comments. We have created new figures (Figures S2 and S3) to address these comments for the organic and aqueous fractions. These additional UpSet plots illustrate the overlap in chemical classes for the observed metabolites in each sample.

Re: mSystems01225-23R2 (Decoding the chemical language of *Suillus* fungi: genome mining and untargeted metabolomics uncover terpene chemical diversity)

Dear Dr. Paul E. Abraham:

Your manuscript has been accepted, and I am forwarding it to the ASM production staff for publication. Your paper will first be checked to make sure all elements meet the technical requirements. ASM staff will contact you if anything needs to be revised before copyediting and production can begin. Otherwise, you will be notified when your proofs are ready to be viewed.

Cover Image Submissions: If you would like to submit a potential Featured Image, please email a file and a short legend to msystems@asmusa.org. Please note that we can only consider images that (i) the authors created or own and (ii) have not been previously published. By submitting, you agree that the image can be used under the same terms as the published article. Image File requirements: TIF/EPS, 7.5 inches wide by 8.25 inches tall (at least 2,250 pixels wide by 2,475 pixels tall), minimum 300 dpi resolution (600 dpi preferred), RGB, and no figure elements, e.g., arrows or panel labels. The legend should be a short description of the image, 1-2 sentences recommended.

Sincerely,
Yu-Liang Yang
Editor
mSystems